

# Are women with endometriosis more likely to experience reduced physical performance compared to women without the condition?

Tatiana Silva[1], Maiara Oliveira[1], Edwiges Oliveira[2], Rayllanne Macena[2], Gessica Taynara de Oliveira Silva[2], Saionara M. A. Câmara[1] and Maria Micussi[1]

[1] Programa de Pós-Graduação em Fisioterapia, Universidade Federal do Rio Grande do Norte, Natal, Rio Grande do Norte, Brazil
[2] Departamento de Fisioterapia, Universidade Federal do Rio Grande do Norte, Natal, Rio Grande do Norte, Brazil

Corresponding author
Maria Micussi,
thereza.micussi@ufrn.br

## ABSTRACT

**Background:** Endometriosis is a condition of the female reproductive system associated with pelvic pain. Chronic pain can affect physical performance by limiting the functional activities, thus, it is hypothesized that women with endometriosis may also present decreased functional capacity, decreased strength, and mobility. The objective of this study is to compare physical performance in women with and without endometriosis.
**Methods:** This is a cross-sectional study composed of 115 women equally divided into two groups: the endometriosis group (EG), composed of women with a confirmed diagnosis of the disease by magnetic resonance imaging, and the comparator group (CG), consisting of women without suspicion of the disease. Physical performance (dependent variable) was assessed using hand dynamometry, the 6-min walk test (6MWT), gait speed, and the chair stands test. CG participants performed the tests during the luteal phase of the menstrual cycle. Descriptive statistics, unpaired t-tests, and chi-square tests were used to describe and compare the groups. Multiple linear regression tested the associations adjusted for covariates (age, income, education, age at menarche, and body mass index).
**Results:** The EG had worse gait speed (mean difference: −0.11; 95% CI: [−0.18 to −0.04]), weaker grip strength (mean difference: −3.32; 95% CI: [−5.30 to −1.33]), shorter distance covered in the 6MWT (mean difference: −83.46; 95% CI: [−121.38 to −45.53]), and a lower number of repetitions in the chair stands test (mean difference: −8.44; 95% CI: [−10.64 to −6.25]) than the CG, even after adjusting for covariates.
**Conclusion:** Grip strength, lower limb strength, mobility, and functional capacity were worse in women diagnosed with endometriosis. Women with endometriosis should be encouraged to engage in physical exercise, adopt healthy lifestyle habits, and participate in rehabilitation activities to control pain, with the aim of reducing functional impairments.

## INTRODUCTION

Endometriosis is a condition of the female reproductive system that can manifest in women since adolescence (*Bulletti et al., 2010*; *Hirsch et al., 2020*). It affects 6–10% of women of reproductive age, of whom 35–50% present with painful symptoms in the pelvic region (*Bulletti et al., 2010*). The prevalence of endometriosis is also elevated in adolescents experiencing pelvic pain (*Hirsch et al., 2020*). A recent survey involving 2,240 adolescents (13–18 years old) revealed that 63% experience distressing menstrual symptoms, resulting in a significant 27% reporting occasional or regular school absences due to these symptoms (*Bush et al., 2017*). It is defined as a chronic, inflammatory, estrogen-dependent pathology characterized by the presence of ectopic endometrial implants and stroma outside the peritoneal cavity, including ovaries, vagina, bladder, peritoneum, gastrointestinal tract, rectum, and lymph nodes (*Taylor, Kotlyar & Flores, 2021*; *Vercellini et al., 2007*). For this reason, endometriosis has been considered a systemic disease and not just a pelvic one (*Taylor, Kotlyar & Flores, 2021*; *Burney & Giudice, 2012*).

Although it may present asymptomatically in some women (*Burney & Giudice, 2012*), the disease may be associated with the presence of symptoms such as dysmenorrhea, deep dyspareunia, dysuria, chronic pelvic pain, and irregular uterine bleeding (*Taylor, Kotlyar & Flores, 2021*; *Vercellini et al., 2007*; *Burney & Giudice, 2012*). A common symptoms includes the sensation of weight in the lumbosacral region of the spine and lower limbs (*Bulletti et al., 2010*), but it can also affect other body parts, being associated with headaches and temporomandibular joint pain (*Wójcik et al., 2023*). These symptoms characterize the disease as a debilitating condition that affects daily life, including professional and personal activities (*Hirsch et al., 2020*; *Vercellini et al., 2007*). In addition, pain can impair social relationships and generate mood changes (*Burney & Giudice, 2012*; *Wójcik et al., 2023*; *Della Corte et al., 2020*).

The repeated experience of intense pain has the potential to alter pain processing systems in the brain (*Grundström et al., 2019*). In a recent study, women with endometriosis presented altered pain thresholds, which are indicative of central sensitization (*Grundström et al., 2019*). Previous research on chronic musculoskeletal pain conditions found a relationship between symptoms indicative of central sensitization and greater functional disability in activities of daily living (*Tanaka et al., 2019*; *Sakulsriprasert, Vachalathiti & Kingcha, 2021*).

Chronic pain can affect physical performance by limiting the functional activities, which also contributes to other symptoms such as anxiety and depression (*Tanaka et al., 2019*; *Sakulsriprasert, Vachalathiti & Kingcha, 2021*). The interplay between chronic pain, anxiety, and depression can further worsen the decline in physical function (*Tanaka et al., 2019*; *Sakulsriprasert, Vachalathiti & Kingcha, 2021*; *Matheve et al., 2022*). In addition, impairment in the physical performance of individuals with chronic pain may include modifications in their ability to perform physical tasks, including aerobic activities, strength, and flexibility (*Taylor, Kotlyar & Flores, 2021*; *Della Corte et al., 2020*; *Tanaka et al., 2019*; *Sakulsriprasert, Vachalathiti & Kingcha, 2021*; *Matheve et al., 2022*; *Evans et al., 2021*). Physical performance can be affected, especially due to pain that hinders the

performance of habitual activities (*Tanaka et al., 2019*; *Sakulsriprasert, Vachalathiti & Kingcha, 2021*; *Matheve et al., 2022*; *Evans et al., 2021*).

Chronic diseases, such as low back pain, and their impact on physical impairment have been extensively studied and demonstrated to be relevant, particularly with regards to decreased gait speed and lower limbs muscle strenght (*Perera et al., 2006*; *Sallinen et al., 2010*). However, the relationships between endometriosis and measures of physical performance have not been the focus of previous studies. It is hypothesized that women with endometriosis may present worse physical performance in terms of functional capacity, decreased strength, and mobility compared to those without the condition. Understanding the impact of endometriosis on physical performance can help identify at-risk groups for disability and guide prevention and rehabilitation strategies. Therefore, the objective of this study is to compare physical performance in women with and without endometriosis.

## METHODS

A cross-sectional study was conducted at the outpatient clinic of the Januário Cicco School Maternity Hospital (MEJC) at the Federal University of Rio Grande do Norte (UFRN), Brazil, following the Strengthening the Reporting of Observational Studies in Epidemiology (STROBE) checklist (*Malta et al., 2010*).

### Sample

The study sample comprised users of the MEJC/UFRN outpatient clinic attended from September 2022 to March 2023. The sample was equally divided into two groups: endometriosis group (EG) and comparator group (CG). The EG included participants seeking service for endometriosis-related care, while the CG consisted of participants seeking various forms of care, such as general gynecology, family planning, and mastology. Due to the lower number of people with endometriosis in the service, for each evaluation of a person in the EG, the evaluation of a person in the CG was also conducted, forming both groups simultaneously during the collection period.

Both groups included women between 20 and 40 years old (EG: 33.12 ± 6.51; CG: 27.51 ± 6.64), with cognitive ability to understand and answer the questionnaires, as observed by the evaluator. Specific inclusion criteria for the endometriosis group were having a diagnosis of endometriosis identified by clinical findings and confirmed by the presence of deep infiltration *via* magnetic resonance imaging (MRI), irrespective of the time of diagnosis. For the CG, the inclusion criteria were women with regular menstrual cycles and in the late luteal phase for physical performance tests. The presence of physical disability that could limit the performance of physical tests ($n = 1$), acute disease of the urinary and/or gynecological tract ($n = 11$), and suspected pregnancy ($n = 1$) were considered exclusion criteria. Of the 128 women recruited for the study, 13 were excluded. The final sample of the study was, therefore, 115 participants.

## Procedures

The women who met the eligibility criteria underwent a standardized evaluation as described below.

## Outcome variables

The dependent variables defined were handgrip strength, 6-min walk test (6MWT), gait speed, and chair-stands test. These tests are widely used in clinical research and practice to validate the functional capacity of individuals.

Handgrip strength was evaluated using a Crown® manual dynamometer according to the Bohannon protocol (*Bohannon, 1986*). Participants were seated in a chair with their feet and trunk supported, shoulders adducted, elbows flexed at 90°, forearms in a neutral position, and wrists extended from 0° to 30°. Participants were instructed to perform a maximal isometric contraction for 5 s, and the peak force was recorded. Three evaluations were performed with a 30 s interval between each repetition, and the mean of the three measures in kilograms was used for analysis (*Rikli & Jones, 2013*). If the examiner recognized any compensatory movements, a new measurement was taken and recorded (*Bohannon, 1986*).

The 6MWT is widely used to evaluate submaximal functional capacity by measuring the maximum distance walked by the individual during 6 min (*ATS Committee on Proficiency Standards for Clinical Pulmonary Function Laboratories, 2002*). Two standardized encouraging phrases were used during the test at the third and fifth minute to increase the participant's physical performance. The distance walked in meters was used as the evaluation measure.

For gait speed evaluation, an eight-foot space was marked with adhesive tape, and the participant was asked to walk from the starting mark to the end mark at their habitual pace. The examiner demonstrated initially and remained beside the participant during the test (*Guralnik et al., 1994*). The time was recorded for two attempts, and the shorter time was used to calculate the gait speed in meters per second (*Guralnik et al., 1994*).

The chair-stands test was also performed for 30 s (*Rikli & Jones, 2013*). This test aims to measure lower limbs strength. A chair with a backrest and seat height of 43 cm from the ground and a stopwatch were used. The participant was asked to sit with their feet fully supported on the ground and arms crossed over their chest. At the signal "start," the participant had to stand up fully from the chair and return to the fully seated position many times as she can. The number of times the person completed the full movement in 30 s was used for analysis (*Rikli & Jones, 2013*).

The age, household income, education, age at menarche, body mass index (BMI, Armonk, NY, USA), and physical activity were considered potential confounders based on the literature. With the exception of BMI, all other variables were self-reported and recorded in the standardized assessment form.

Age, household income, and education were considered possible confounding factors as they are associated with physical performance (*Kuh et al., 2005*). Household income was categorized based on the monthly minimum wage (MW), defined as the lowest remuneration that employers can legally pay to workers. Theoretically, this minimum

value should cover a family's normal needs for food, housing, clothing, hygiene, and transportation. At the time of the study, the MW was set at R$1,220.00 (one thousand two hundred and twenty Brazilian reais). In our study, household income was dichotomized into up to 1 MW and more than 1 MW. Regarding education, the variable was categorized into up to 8 years of schooling and more than 8 years of schooling. Age in years was analyzed as a continuous variable.

The age at menarche may be related to factors such as physical activity, nutrition, and overall health of women, which may influence physical performance (*Abreu-Sánchez et al., 2020*). Additionally, menarche marks the beginning of increased sex hormone levels, making adolescent girls/women more susceptible to hormonal fluctuations and their consequences. For data analysis, age at menarche in years was used as a continuous variable.

The body mass index (BMI, Armonk, NY, USA) is associated with better muscle strength, worse functional capacity (*Le Strat et al., 2020*; *Hergenroeder et al., 2011*), and poorer performance in activities such as gait speed and chair-stands. The BMI (kg/m2) was calculated from measured height (m) and weight (kg) and subsequently categorized according to the World Health Organization (WHO) international classification as: 18.5 to 24.99 (normal weight), 25.00 to 29.99 (overweight), 30.00 to 34.99 (obese I), ≥35.00 (obese II and III).19 In this study, the BMI was categorized as up to 24.99 kg/m2 (normal) and above 25.00 kg/m2 (overweight and obese) (*Ministério da Saúde, 2007*).

Physical activity and exercise promote benefits for the body through muscle contraction that releases myokines(26). Myokines can exert direct effects on the muscle themselves or on distal organs (*Brandt & Pedersen, 2010*). Furthermore, exercise stimulates the production of leukocytes, cortisol, and adrenaline, which also have acute anti-inflammatory effects (*Nimmo et al., 2013*). Physical activity was categorized as either "more active" (report of physical exercise practice above 150 min per week) and "less active" (less than 150 min per week) (*Haskell et al., 2007*).

To characterize the sample, other variables were collected, including medication use and assessment of pain symptomatology using the Brief Pain Inventory (BPI). This questionnaire is a multidimensional instrument that employs a 0–10 scale to rate the following items: pain intensity, interference of pain on the ability to walk, impact on the patient's daily activities, work, social activities, mood, and sleep. The pain reported by the patient encompasses the one experienced at the time of the completing the questionnaire, as well as the most intense, least intense, and the average pain over the last 24 h (*Bennett, 2009*). The inventory is already validated and adapted to the Portuguese language, consisting of 10 questions that make up two scales, the pain intensity and severity scale (questions 3, 4, 5, 6) and the pain interference scale in the individual's life (question 9 a–g). The final score of each scale ranges from 0–10, with 0 indicating no pain and 10 representing the worst possible pain (*Bennett, 2009*).

## Statistical analysis

The statistical analysis was performed using SPSS version 21.0 (IBM Corp., Armonk, NY, USA) (*BM Corp, 2011*). Sociodemographic and clinical data were described by means,

medians, and standard deviations for continuous variables, and absolute and relative frequencies for categorical variables. To compare the groups with and without endometriosis, the unpaired Student's t-test was used for continuous variables and the Chi-square test for categorical variables. Multiple linear regression models were applied to assess the difference between the two groups in each of the functional performance tests, adjusted for potential covariates of clinical relevance (age, family income, education, physical activity, menarche, and BMI). The models were checked regarding the assumptions of normality of errors, linearity, homoscedasticity, and absence of multicollinearity. The level of significance was defined as a $p$-value < 0.05.

### Ethical aspects

The study was conducted in accordance with the Helsinki Declaration. All participants were briefed on the research procedures, and those who agreed signed the consent form. The project was submitted and approved by the local Ethics Committee of the Federal University of Rio Grande do Norte (approval number: 5.695.435).

## RESULTS

The characteristics of the sample according to the diagnosis of endometriosis are presented in Table 1. All participants in the EG were receiving pharmacological treatment for endometriosis (dienogest), and a mean of 6.14 ± 2.14 was observed for pain evaluation. All physical performance tests showed statistically significant differences between the groups, with the EG exhibiting worse results. Women in the EG were older, had a higher proportion of women who reported having a partner, who were less physically active, and who were overweight or obese, compared to the CG. There were no significant differences between the groups in terms of education level, family income, and age at menarche.

Table 2 presents unadjusted analyses for physical performance tests according to the covariates. Women without a partner showed greater strength. Participants classified as overweight (BMI ≥ 25 kg/m2) presented worse performance in all tests compared to those with normal BMI.

Table 3 shows the results of multiple linear regression for each performance test. Participants with endometriosis performed fewer repetitions in the chair-stands test, covered a shorter distance in the 6 MWT, presented slower gait speed, and had weaker handgrip strength than the group without endometriosis, even after adjusting for covariates (Fig. 1). These results confirm the study hypothesis.

## DISCUSSION

This study compared the physical performance, assessed through strength and mobility tests, of women with and without endometriosis. Our results show that endometriosis is associated with worse physical performance in all tests considered, even after controlling by potential confounders.

In the same study, a longer duration of endometriosis pain was associated with increased sensitization (Evans et al., 2021).

**Table 1 Characteristics of the sample according to sociodemographic data, clinical variables, and physical performance measures ($N = 115$).**

| Variables | Endometriosis group ($n = 54$) | Comparator group ($n = 61$) | p-value |
|---|---|---|---|
| Age (years) | 31.00 ± 4.71 | 25.49 ± 5.46 | <0.001[a] |
| Education (%) | | | 0.29[b] |
| ≤8 years of education/no response | 37 (68.50) | 36 (59.00) | |
| ≥9 years of education | 17 (31.50) | 25 (41.00) | |
| Household income (%) | | | 0.53[b] |
| ≤1 MW/No response | 18 (33.30) | 17 (27.90) | |
| ≥2 MW | 36 (66.70) | 44 (72.10) | |
| Age at menarche (years) | 12.34 ± 1.18 | 12.00 ± 1.28 | 0.15[b] |
| BPI | 6.40 (4.75–7.70) | 1.00 (0.00–2.40) | <0.001[c] |
| BMI (kg/cm$^2$) | | | 0.008[b] |
| ≤24.9 | 22 (40.70) | 40 (65.60) | |
| ≥25 | 32 (59.30) | 21 (34.40) | |
| Physical activity (%) | | | 0.04[b] |
| Less active | 40 (74.10) | 34 (55.70) | |
| More active | 14 (25.90) | 27 (44.30) | |
| Physical performance | | | |
| chair-stands (n) | 15.26 ± 5.22 | 24.51 ± 5.36 | <0.001[b] |
| 6MWT (m) | 405.65 ± 108.49 | 489.82 ± 73.02 | <0.001[b] |
| Gait speed (m/s) | 0.82 ± 0.18 | 0.96 ± 0.17 | <0.001[b] |
| Handgrip strength (KgF) | 21.72 ± 5.54 | 25.82 ± 4.02 | <0.001[b] |

**Notes:**
Categorical data are presented as absolute frequency (n) and relative frequency (%), while continuous data are presented as mean and standard deviation. Higher values represent better physical performance in all physical performance tests. n, number; 6MWT, 6-min walk test; m, meters; m/s, meters per second; MW, minimum wage; BPI, brief pain inventory; BMI, body mass index; Kg/cm$^2$, kilogram per square centimeter; KgF, kilogram-force.
[a] *p*-value for Student's t-test for independent samples.
[b] *p*-value for Chi-square test.
[c] *p*-value for Mann-Whitney test.

Understandably, women with endometriosis in our study reported higher levels of pain compared to those without endometriosis. A recent study demonstrated that women with endometriosis are more likely to experience widespread pain, fatigue, and fatigue-related disability in daily activities, as measured by the fatigue questionnaire, than women without endometriosis (*Evans et al., 2021*). The repetitive experience of intense pain in endometriosis may be one of the factors explaining the worse physical performance in this group. It is known that endometriosis has the potential to alter pain processing systems in the brain, and in a recent study, women with suspected endometriosis showed altered pain thresholds indicative of central sensitization (*Evans et al., 2021*). In the same study, a longer duration of endometriosis pain was associated with increased sensitization (*Evans et al., 2021*). Besides pain, psychological stress may further compound these negative effects, potentially amplifying the impact on crucial life tasks such as leisure and work. This reduction in the level of physical activity could, consequently, lead to a decline in components essential for physical performance, such as muscle strength and mobility.

**Table 2 Mean physical performance according to covariates ($n$ = 115).**

|  | Chair-stands (n) | 6MWT (m) | Gait speed (m/s) | Handgrip strength (KgF) |
|---|---|---|---|---|
| **Education** |  |  |  |  |
| ≤8 years of education/no response | 20.10 ± 6.73 | 446.85 ± 107.81 | 0.89 ± 0.17 | 23.26 ± 5.19 |
| ≥9 years of education | 20.29 ± 7.58 | 457.51 ± 85.59 | 0.90 ± 0.20 | 24.99 ± 5.08 |
| $p$-value[a] | 0.89 | 0.59 | 0.98 | 0.09 |
| **Household income** |  |  |  |  |
| ≤1 MW/ No response | 20.60 ± 7.46 | 456.39 ± 97.59 | 0.91 ± 0.20 | 22.89 ± 6.80 |
| ≥2 MW | 19.98 ± 6.86 | 448.26 ± 101.69 | 0.89 ± 0.18 | 24.33 ± 4.29 |
| $p$-value[a] | 0.66 | 0.69 | 0.70 | 0.17 |
| **BMI (kg/cm$^2$)** |  |  |  |  |
| ≤24.9 | 21.97 ± 7.44 | 468.52 ± 106.33 | 0.92 ± 0.19 | 25.01 ± 5.07 |
| ≥25 | 18.06 ± 5.88 | 429.43 ± 88.56 | 0.86 ± 0.16 | 22.59 ± 5.08 |
| $p$-value[a] | <0.01 | 0.03 | 0.04 | 0.01 |

Notes:
Higher values on the tests indicate better physical performance. n, number; 6MWT, 6-min walk test; m, meters, m/s, meters per second; MW, minimum wage; BMI, body mass index; Kg/cm$^2$, kilogram per square centimeter; KgF, kilogram-force.
[a] $p$-value is for the $t$-test for independent samples.

**Table 3 Multiple regression models for the functional capacity tests adjusted for the covariates.**

|  | Mean difference | CI 95% | $p$-value |
|---|---|---|---|
| **Chair-stands** |  |  |  |
| Endometriosis | −8.44 | [−10.64 to −6.25] | <0.001* |
| Control | 0 |  |  |
| **6MWT** |  |  |  |
| Endometriosis | −83.46 | [−121.38 to −45.53] | <0.001* |
| Control | 0 |  |  |
| **Gait speed** |  |  |  |
| Endometriosis | −0.11 | [−0.18 to −0.04] | 0.003* |
| Control | 0 |  |  |
| **Handgrip strength** |  |  |  |
| Endometriosis | −3.32 | [−5.30 to −1.33] | 0.001* |
| Control | 0 |  |  |

Notes:
Models adjusted for age, household income, education, physical activity, age at menarche, and body mass index. CI, confidence interval. 6MWT, 6-min walk test.
* Statistically significant: $p < 0.05$.

The performance of the gait speed test, 6MWT, handgrip strength, and chair-stands test has not been evaluated in women with endometriosis in previous studies. However, similar to our results, worse physical performance has been reported in other populations with a diagnosis of untreated chronic pain (*Rodrigues et al., 2017*). Chronic pain can induce fear and movement blocking behaviors, which often results in low physical performance (*Luque-Suarez, Martinez-Calderon & Falla, 2019*).

Previous studies have investigated physical performance in women with chronic pain using the 6MWT. One study showed that women with chronic pain presented significantly

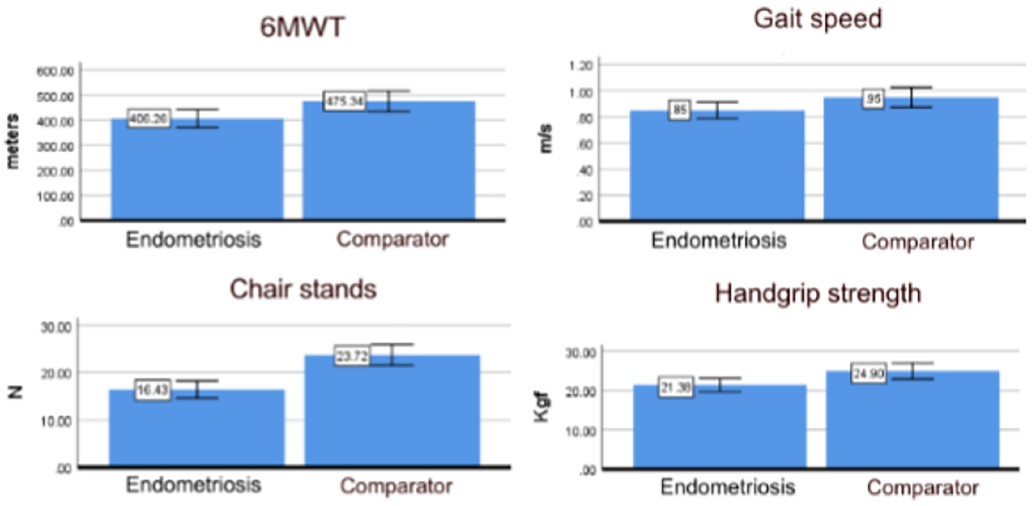

**Figure 1 Physical performance results (mean and 95% CI) for endometriosis and comparator groups estimated by the linear regression analysis adjusted for age, education, income, age at menarche and BMI ($N$ = 115).** A visual representation of the results presented in Table 3 considering a participant aged 28.7 years, with more than 9 years of school, income above two minimum wages, menarche at 12.1 years, and BMI over 25 Kg/m$^2$.

worse results in the 6MWT when compared to those without pain (335 *vs* 680 m), indicating a reduction in physical activity capacity (*Bennett, 2009*). Another study found poorer physical performance in the 6MWT when comparing women with fibromyalgia diagnosis (441.8 ± 84.1 m) to healthy women (523.9 ± 80.3 m) (*Rodrigues et al., 2017*; *Breda et al., 2013*).

In our findings, women with endometriosis walked an average of 80 m less than the women without the disease (*Rodrigues et al., 2017*; *Breda et al., 2013*). This difference is greater than clinically relevant change for the 6 MWT, indicating worse functional capacity associated with endometriosis (*Breda et al., 2013*; *Nordeman et al., 2017*). The mean distance walked by the endometriosis group was 443.50 m, which is similar to the results found in previous studies (*Rodrigues et al., 2017*; *Breda et al., 2013*) with patients with chronic disease patients. It is known that the 6MWT can predict physical disability and indicate a prognosis for accelerating disability. A 2-year cohort study showed that low 6MWT values represent 54% of the reduction in day life activity in individuals with chronic low back pain during their lifetime (*Perera et al., 2006*).

Both the chair-stands test and gait speed are functional measures that assess the lower limbs strength and mobility, respectively. Our results showed that women with endometriosis, on average, had slower gait speed by 0.11 m/s, representing a clinically relevant difference (*Nordeman et al., 2017*). They also performed almost nine repetitions less on the chair-stands test than women without endometriosis diagnosis, indicating much worse performance. These tests represent simple tasks commonly performed in daily life that, however, involve complex actions between musculoskeletal and neuromotor components. Both tests are widely used in clinical practice and research as they are valuable for identifying people at risk of disability (*Nordeman et al., 2017*). Despite this, to

the best of our knowledge, this is the first study that evaluated performance on these tests in women with endometriosis.

Handgrip strength results indicated an average of approximately 3 KgF lower strength in women with endometriosis (*Breda et al., 2013*). The lower strength can be attributed not only to pain, which can negatively affect the ability to generate force (*Tanaka et al., 2019*; *Sakulsriprasert, Vachalathiti & Kingcha, 2021*) but also to hormonal factors. In our study, all participants with endometriosis were receiving drug treatment with substances that reduce estrogen levels, as the disease is known to be estrogen-dependent. Previous studies have shown a positive association between estrogen levels and better strength/muscle function (*Breda et al., 2013*; *Skelton et al., 1999*). Considering this relationship between sex hormones and muscle action, we chose to perform physical performance tests in the comparator group during the late luteal phase when estrogen levels are naturally lower.

The predictive ability of handgrip strength in identifying adverse outcomes such as disability and mortality has already been widely demonstrated among the elderly populations (*Sallinen et al., 2010*; *Rodrigues et al., 2017*; *Cooper et al., 2011*). For older women, a cutoff point of 21 kg in handgrip strength has been used as an indicator of a higher probability of mobility disability (*Sallinen et al., 2010*; *Cooper et al., 2011*). In adult women within the age range of the present study participants, a cutoff point of 31 kg has been indicated as a threshold for reduction of functionality (*Sallinen et al., 2010*). The fact that the average strength of the sample with endometriosis in this study is below this threshold indicates that this group is at greater risk of experiencing disability at younger ages. This can significantly impact the quality of life of these women, especially as they age, when there is expected to be even greater reduction in muscle strength due to the aging process itself.

The fact that young women with endometriosis have already worse results in all evaluated physical performance tests demonstrates the importance of including physical performance evaluation in this population. This will allow the monitoring of changes in functional performance over time and identification of those who would benefit from rehabilitation and disability prevention strategies. Procedures for pain control, combined with encouragement to engage in physical activity, can help mitigate the negative effects of endometriosis on women's physical health. The inclusion of fatigue assessment is recommended in future investigations on endometriosis, considering the possible occurrence of this symptom as a consequence of pain, as well as directly from the disease, which can significantly impair patients' physical performance.

## Strength and limitation

According to our literature search, this is the first study that investigates the association between physical performance and endometriosis. Objective tests were used to evaluate physical performance, which allows for accurate comparisons and minimizes measurement bias. The cross-sectional design of the study limits causal inferences and the results cannot guarantee the development of disability over the years.

All participants were recruited from a specialized service for endometriosis, which suggests that they may represent a population with more severe pain symptoms than the

general endometriosis population. Moreover, in the region where the study was conducted, there is a scarcity of healthcare services, including limited access to diagnostic imaging such as MRI. This may result in differences in sociodemographic and clinical characteristics between the groups and could contribute to delayed diagnosis, thereby hinder early treatment. It is plausible that women with asymptomatic endometriosis, as well as those in the early stages of the disease, were not represented in the sample. As previously mentioned, pain and psychological stress are possible intermediators in the associations we found. Future studies aiming to understand the factors in the pathway between endometriosis and worse physical performance should assess the presence and intensity of these symptoms.

## CONCLUSIONS

This study confirms the hypothesis that women diagnosed with endometriosis present worse physical performance. Handgrip strength, lower limbs strength, mobility, and functional capacity were all found to be diminished in women diagnosed with endometriosis. Clinical strategies should be developed with the aim of improving physical performance and quality of life. Therefore, it is suggested that women with endometriosis should be encouraged to engage in physical exercise, adopt healthy lifestyle habits, and participate in activities aiming at pain control.

### Funding
The APC for this article was funded by the Coordenação de Aperfeiçoamento de Pessoal de Nível Superior (CAPES), Finance Code 001.

### Grant Disclosures
The following grant information was disclosed by the authors:
Coordenação de Aperfeiçoamento de Pessoal de Nível Superior (CAPES): 001.

### Competing Interests
Maria Micussi is an employee of the Federal University of Rio Grande do Norte, Brazil, RN/Natal.

### Author Contributions
- Tatiana Silva conceived and designed the experiments, prepared figures and/or tables, and approved the final draft.
- Maiara Oliveira performed the experiments, prepared figures and/or tables, and approved the final draft.
- Edwiges Oliveira performed the experiments, authored or reviewed drafts of the article, and approved the final draft.
- Rayllanne Macena performed the experiments, authored or reviewed drafts of the article, and approved the final draft.

- Gessica Taynara de Oliveira Silva performed the experiments, authored or reviewed drafts of the article, and approved the final draft.
- Saionara M. A. Câmara conceived and designed the experiments, analyzed the data, authored or reviewed drafts of the article, and approved the final draft.
- Maria Micussi conceived and designed the experiments, analyzed the data, authored or reviewed drafts of the article, and approved the final draft.

### Human Ethics

The following information was supplied relating to ethical approvals (*i.e.*, approving body and any reference numbers):

The Federal University of Rio Grande do Norte granted Ethical approval to carry out the study within its facilities (Ethical Application Ref: 5.695.435).

### Data Availability

The raw data are available in the Supplemental File.

### Supplemental Information

Supplemental information for this article can be found online at http://dx.doi.org/10.7717/peerj.16835#supplemental-information.

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
