# Peer review of "Are women with endometriosis more likely to experience reduced physical performance compared to women without the condition?"

_PeerJ, doi:10.7717/peerj.16835_

## Round 0.1 · original submission · Minor Revisions

Dear authors,

According to reviewers, the manuscript should be improved. The research hypothesis should be stated and other confirmations reviewed on Introduction section. More clarify is suggested on Experimental design section. In addition, the conclusions are must be improved as well.

Thus, please, we are waiting your comments and review to continue with the process.

Regards

Reviewer 1 ·

Basic reporting

1. Introduction
line 114 - the authors write about the occurrence of pain in the body, it is worth adding that pain can occur in a different part of the body than the pelvis-or L-S segment-for example, in the temporomandibular joints. I propose to refer to: https://doi.org/10.3390/jcm12082862

**Staff Note: Regarding the above suggestion for citation, it is PeerJ policy that additional references suggested during the peer-review process should only be included if the authors are in agreement that they are relevant and useful **

line 134 - the research hypothesis should also be stated and the results should describe whether the study confirmed it.
2. Sample
line 153 - necessarily characterise the study group by giving their age as the mean and standard deviation, how many years they have suffered from endometriosis, i.e. since the medical diagnosis,
line 161 - please indicate clearly the inclusion and exclusion criteria for participation in the study.

Experimental design

3. Statistical analysis
line 240 - The authors used SPSS version 21.0 (IBM Corp., Armonk, NY, USA) - please provide a refrence e.g. article/book/website to this tool.
4. Results - the authors have included the results in tables, could the authors prepare 1-2 figures? the results would then be more interesting for the reader.
The research hypothesis should also be stated and the results should describe whether the study confirmed it – point 1.

Validity of the findings

5. Conclusion - The conclusion should be a response to a research hypothesis that is not there.
Necessarily, in the introduction, the authors must add the research hypothesis and speak to it.

Additional comments

Thank you for the opportunity to review an interesting manuscript.

Annotated reviews are not available for download in order to protect the identity of reviewers who chose to remain anonymous.

·

Basic reporting

Although the language used is easy to understand, the manuscript would benefit from a revision by a scientifically trained fluent English speaker. In my opinion, an error has already appeared in the first line of the abstract. “Background. Endometriosis is a condition of the female reproductive ? associated with…” or in line 130 “musgle strength”.
The manuscript contains a sufficient number of coherent references. The first part of the introduction, in particular the prevalence, could be supported even more by rele-vant literature, e.g. Hirsch, M., Dhillon-Smith, R., Cutner, A. S., Yap, M., & Creighton, S. M. (2020). The prevalence of endometriosis in adolescents with pelvic pain: a sys-tematic review. Journal of Pediatric and Adolescent Gynecology, 33(6), 623-630. The selection of confounding variables is not explained sufficiently enough in the back-ground section. Psychological stress (e.g. depressive symptoms), which plays an enormous role in women affected by endometriosis, is briefly mentioned in the intro-duction, but unfortunately not included in the study.
The structure of the manuscript (introduction, methods section, results, discussion and conclusion) is in itself adequate. Only the subdivision of the methods section (e.g. Dependent variables, covariates, other variables) seems unusual; and adapta-tion to more common subdivisions (e.g. outcome variables, measurements), would be worthwhile. The tables appear professional and clear to me and raw data is pro-vided. Unfortunately, the raw data does not match the information in the manuscript. For example, the maximum age of the test subjects.
Hypotheses had not been formulated.

Experimental design

The study described here investigates whether patients with and without endometrio-sis differ in terms of physical performance. Research in the field of endometriosis is fundamentally necessary and highly relevant, as many women are affected and there are still many unexplained aspects of the disease. This is the first original research and the research question generally corresponds to the aims and scope of the jour-nal.
Unfortunately, the research question is not formulated precisely enough. For exam-ple, it is not clear what role pain plays in the study of subjects with and without en-dometriosis with regard to physical performance.
Objective tests were used to analyze physical performance and ethical standards appear to have been met.
The investigations and statistical analyses carried out are described in a comprehen-sible manner. However, the identification and explanation of the study and obtaining the written consent of the test subjects is missing.

Validity of the findings

All underlying data have been provided. In my opinion, a great weakness of the manuscript is that the two groups were not matched and are therefore not sufficiently comparable. Both groups differ significantly with regard to all variables to be con-trolled. A second shortcoming is that the statistical method used does not fit the re-search question. Multiple linear regression analyses are usually used to examine the relationship between several independent variables and one dependent variable, ra-ther than to compare two groups, as is the case here.
Unfortunately, the conclusions drawn are not consistent. It is discussed that intensity and pain processing could be an underlying mechanism for the difference between the two groups. However, pain was only recorded in one group and is not included in the further statistical analysis.
In my opinion, a better matching of control and patient groups, a more coherent sta-tistical analysis and the inclusion of pain and psychological aspects (e.g. anxiety and depression) are necessary adjustments.

---

## Round 0.2 · accepted · Accept

I am writing to inform you that your manuscript has been Accepted for publication. Congratulations!